# In-person social isolation in the age of smartphones: Examining age, period, cohort effects by gender

Siyun Peng[1,2*], Shawn Bauldry[3]

1 School of Aging Studies, University of South Florida, Tampa, Florida, United States of America,
2 Department of Sociology, Indiana University, Bloomington, Indiana, United States of America,
3 Department of Sociology, Purdue University, West Lafayette, Indiana, United States of America

* siypeng@iu.edu

## Abstract

### Objectives

The rise of smartphones and social media is widely seen as a pivotal societal shift that has fueled what the U.S. Surgeon General in 2023 described as "our epidemic of loneliness and isolation." The evidence for an increase in social isolation, however, has not accounted for age and cohort effects – i.e., variation in time spent alone over the life course and across generations. This study leverages the latest breakthrough in Age-Period-Cohort (APC) modeling to disentangle age-specific effects, societal changes, and generational shifts that contribute to social isolation in the era of smartphones.

### Methods

We analyze data from the 2003–2022 waves of the American Time Use Survey (ATUS), focusing on time spent alone in nonwork activities among individuals aged 15–79 (N = 240,576). Using a recent breakthrough in APC modeling, we identify net effects of age, period, and cohort separately for men and women.

### Results

Social isolation has increased over the past two decades for both women and men, with a notable acceleration in the mid-2010s, suggesting that societal shifts during this period may have intensified isolation. However, age and cohort effects play a much larger role in explaining the social isolation crisis of the 21st century. Gender differences are substantial, particularly in cohort trajectories and in age patterns after age 70.

**Data availability statement:** American Time Use Survey is publicly available via https://www.bls.gov/tus/data.htm.

**Funding:** This work was funded by the National Institute on Aging (R01AG057739; R01AG070931; R01AG078247; R01AG076032). The content is solely the responsibility of the authors and do not necessarily represent the official views of the National Institutes of Health.

**Competing interests:** The authors have declared that no competing interests exist.

## Discussion

While smartphone-era societal shifts have contributed to a general rise in isolation, aging and generational differences explain more of the variation. Public health efforts should prioritize mitigating isolation among older adults or earlier cohorts, with attention to gender-specific patterns.

## Introduction

In 2023, the US Surgeon General, Dr. Vivek Murthy, announced a public call for the Americans' attention to "our epidemic of loneliness and isolation" [1]. In his call, Murthy cites the rise of technologies, such as smartphones, social media, online entertainment, and remote work, as contributors to a social isolation epidemic. Although smartphones offer unprecedented opportunities to stay in touch with friends and family and connect with strangers online [2–5], there is growing concern that they are contributing to a decline in face-to-face interactions [6–8]. As more individuals turn to digital platforms for social engagements and entertainment, the potential for weakened in-person relationships and diminished real-world social support has become a pressing public health issue [1].

Despite the apparent consensus on the social isolation crisis, much of the evidence on this topic fails to account for the potential effects of age and cohort that could account for some of the observed pattern over time (i.e., a period effect) [9]. For example, the U.S. Surgeon General's public report highlights a time-trajectory study using data from the American Time Use Survey (ATUS), which shows an increase in time spent alone from 2003 to 2020, as evidence of rising social isolation [1,10]. However, the observed increase in time spent alone from 2003 to 2020 reflects a combination of age, period, and cohort effects. Although many interpret this as a period effect—Americans across all age groups are becoming increasingly socially isolated due to societal changes (e.g., the rise of technologies like the introduction of the iPhone in 2007)—age and cohort effects may also be contributing factors.

The U-shaped age effect in social isolation is well-documented in the literature, with levels of isolation increasing after the 30s [11–13]. This indicates that older adults tend to experience greater social isolation than younger individuals due to life events such as retirement, becoming empty nesters, and the loss of friends and family [11,12,14]. Although evidence shows that people can maintain a stable number of strong relationships across the life course [15–19], they often experience social decline in overall social interactions with age [11–14]. Consequently, the aging demographics in America might be contributing to the observed rise in social isolation.

There may be a cohort effect, where only younger generations experience higher levels of social isolation due to smartphones, while older generations are less affected. Growing up with smartphones, younger generations may develop norms that reduce face-to-face social interactions. This aligns with Putnam's argument that younger generations are more likely to play video games, watch videos, and interact

with friends online, rather than engaging in face-to-face interactions, compared to older generations [20]. By contrast, Fischer in *Still Connected* finds that while younger generations may exhibit different patterns in their social interactions (i.e., less dinner parties and more online communication), social connectedness with friends and family remained largely the same across all other dimensions (i.e., satisfaction with friendships, frequency of contact) [21].

Without the use of Age-Period-Cohort (APC) models, we are not able to distinguish the net effects of age, period, and cohort. To fill this gap, this study leverages a recent APC modeling breakthrough [22] to examine in-person social isolation among men and women, separating the contributions of age, period, and cohort in the era of widespread smartphone use.

## Method

### Data

Data for this study come from the 2003–2022 waves of the American Time Use Survey (ATUS), which is conducted by the U.S. Census Bureau with funding from the Bureau of Labor Statistics to collect information on how Americans spend their time. The ATUS is a nationally representative sample that draws on a subset of households from the Current Population Survey (CPS) each month. An eligible person from the household (a person of at least 15 years) was randomly selected to be interviewed. Interviews were conducted by telephone using a computer-assisted telephone instrument (CATI). The time diary provided a 24-hour recall from each respondent, starting at 4:00 a.m. the previous day and ending at 4:00 a.m. on the interview day. Interviews usually took place the day after the reference day to ensure the events were still fresh in memory. ATUS collected information about what the respondent was doing, how long each activity lasted, where each activity occurred, and whom the respondent was with. The shortest unit of time reported for a given activity is 5 minutes, which allowed for up to 288 activities on a given day, thus providing a finely-grained portrait of everyday social dynamics. The analysis sample consists of respondents between the ages of 15 and 79, which over the 20-year period results in an N of 240,576 cases.

### Measures

**Social isolation.** The ATUS collects "Who" questions for all activities, except for sleeping and personal activities (e.g., grooming, cuddling, having sex, etc.). "Who" questions are used to determine whether respondents spent face-to-face time with others or alone. Many studies have used time alone as a measure of social isolation [10,23]. We excluded alone time during work-related activities to ensure comparability with the study cited in the U.S. Surgeon General's public call [1,10]. In addition, ATUS did not collect "Who" questions for work-related activities before 2010. To compare social isolation patterns since 2003, we need to exclude alone time at work.

**Age, period, and cohort measures.** Following common practice in age-period-cohort (APC) analyses, we constructed 13 indicators for five-year age groups beginning at age 15 and ending at age 79 (i.e., 15–19, 20–24, …, 75–79). Similarly, we constructed 4 indicators for five-year periods beginning in 2003 and ending in 2022 (i.e., 2003–2007, 2008–2012, …, 2018–2022). Finally, we took the difference between the period and age indicators, which resulted in 16 primarily eight-year cohort groups beginning with the 1924 birth cohort, the oldest in the sample, and ending with the 2007 cohort, the youngest in the sample (i.e., 1924–1932, …, 1999–2007).

**Additional covariates.** Because people's daily social contact may change depending on whether it is a working day or not, indicators of *days of the week* (i.e., Monday to Sunday) and *holidays* (yes or no) are included.

### Analytic method

The well-known challenge for any analysis attempting to identify net age, period, and cohort effects involves the linear dependency between the three components (i.e., Age = Period – Cohort). There have been numerous attempts to address this challenge over the last 50 years [24–26], but in one or form or another they all rely on untestable assumptions. The current state of the art for APC analyses, as proposed by Fosse and Winship [9,22], is based on two features of the these

models: [1] the dependency across the three components is only for linear effects and it is possible to obtain estimates of nonlinear effects for each component and [2] it is possible to estimate the parameters for a canonical solution line that characterizes all of the potential linear age, period, and cohort effects that are consistent with the data. In this study, we applied the Fosse and Winship APC approach [9,22] to address our research questions. We adopt their notation in describing the models and recommendations for illustrating results.

Consider the following linear regression model for time spent alone in nonwork activities:

$$y_i = \beta_0 + \alpha_L A_{iL} + \pi_L P_{iL} + \gamma_L C_{iL} + \alpha'_{NL} \mathbf{A}_i + \pi'_{NL} \mathbf{P}_i + \gamma'_{NL} \mathbf{C}_i + \beta'_{NL} \mathbf{x}_i + \epsilon_i, \tag{1}$$

where $i$ indexes respondents, $y$ represents minutes spent alone in nonwork activities, $\mathbf{A}$, $\mathbf{P}$, $\mathbf{C}$ are vectors of indicators for age, period, and cohort groups, $\alpha$, $\pi$, and $\gamma$ are coefficients for the age, period, and cohort indicators, $\mathbf{x}$ is a vector of additional covariates (days of the week and holidays), and the subscript $L$ refers to linear components, while the subscript $NL$ refers to nonlinear components.

In this form of the model, the linear components of the age, period, and cohort effects ($A_{rL}$, $P_{rL}$, $C_{rL}$) are separated from the nonlinear components. This separation is made possible with orthogonal polynomial contrasts rather than the standard effect coding for the indicator variables [27]. Given the unequal number of cases within each age, period, and cohort group, weighted orthogonal contrasts are needed to maintain the independence of the linear and nonlinear effects [28].

The linear regression model in equation (1), however, remains inestimable due to the dependency among the linear age, period, and cohort effects. The two following quantities, which can be used to define a canonical solution line, are estimable.

$$\theta_1 = \alpha_L + \pi_L \tag{2}$$

$$\theta_2 = \gamma_L + \pi_L. \tag{3}$$

With these quantities, the model given in equation (1) can be rewritten as

$$y_i = \beta_0 + \theta_1 A_{iL} + \theta_2 C_{iL} + \alpha'_{NL} \mathbf{A}_i + \pi'_{NL} \mathbf{P}_i + \gamma'_{NL} \mathbf{C}_i + \beta'_{NL} \mathbf{x}_i + \epsilon_i, \tag{4}$$

and fit to the ATUS data. Estimates for $\theta_1$ and $\theta_2$ can then be used in what Fosse and Winship [22] refer to as a 2-D APC plot to examine the range of possible linear age, period, and cohort effects consistent with the data (see Fig 2).

The canonical solution line represents all possible sets of linear age, period, and cohort effects that are consistent with the data. In some cases, it is possible to place bounds on the linear effects that reflect substantive knowledge about the context for the analysis. With bounds placed on the linear effects, one can use estimates from Equation (4) to calculate a range of net total age, period, and cohort effects reflecting a combination of the linear effects and the nonlinear deviations from the linear effects (see Fig 3).

Analyses are stratified by gender. Weights are applied to all models to adjust for non-response and sampling design [29]. All models include the age, period, and cohort indicators along with indicators for days of the week and holidays. Orthogonal polynomial contrasts beyond 5-order polynomials were set to 0 to smooth the estimates [28]. All analyses were conducted in R. Replication code is available online at [redacted for review].

## Results

Fig 1 illustrates age, period, and cohort averages in time spent alone in nonwork activities from 2003 to 2022 for Americans ages 15–79 born between 1924 and 2007. During this period, as emphasized in the Surgeon General's report, the average time spent alone in non-work activities increased from 268 minutes for men and 282 minutes for women in

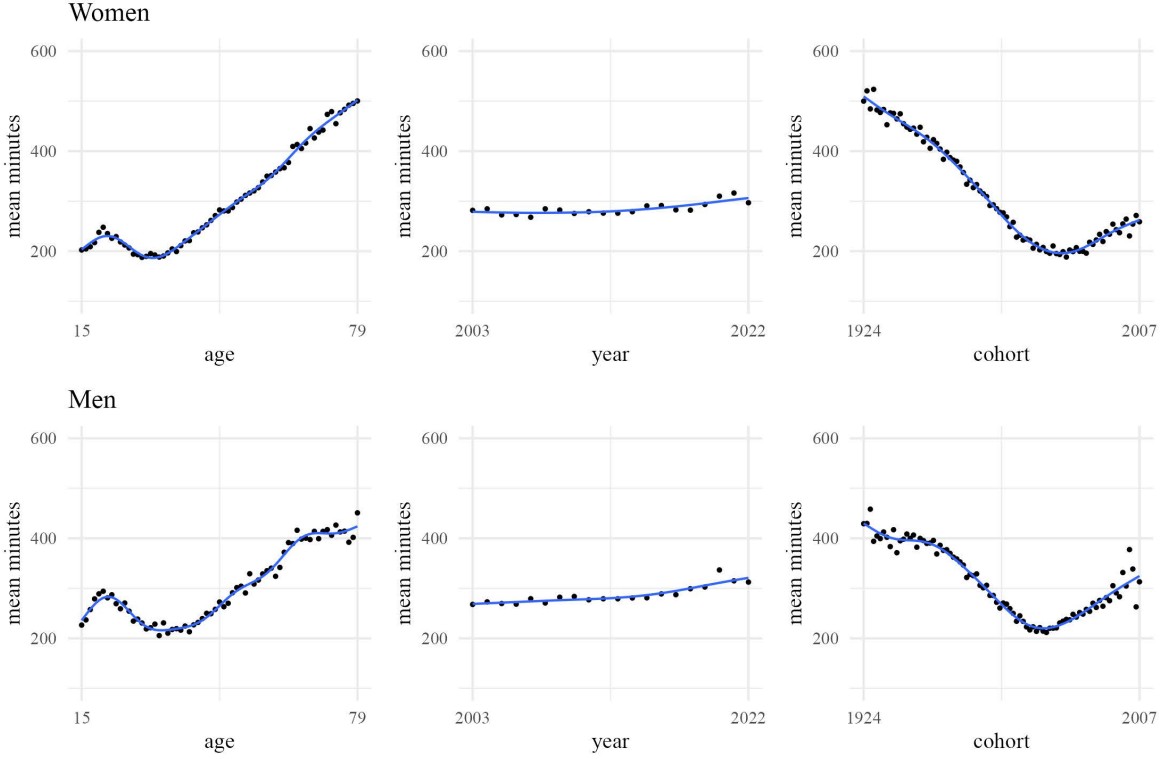

**Fig 1. Average minutes spent alone in nonwork activities by age, period, and cohort for women and men. Estimates of means at each age, year, or cohort apply the ATUS 2003–22 sample weights. Non-parametric lines based on Generalized Additive Models.**

2003–312 minutes for men and 297 minutes for women in 2022 with a slight uptick in the pattern around 2013 or 2014. Turning to age, we see an initial increase in women's average minutes spent alone from ages 15–20, followed by a decrease to a minimum of 188 minutes at age 33, and then a steady rise to 500 minutes by age 79. Men follow a similar pattern up to age 65: an increase from 15 to 20, a decrease to a minimum of 205 minutes at age 33, and then a rise to 416 minutes by age 65. After 65, the increase slows, reaching only 451 minutes by age 79, forming an almost flat plateau. Overall, these age effects align with well-documented life course processes in social interactions, with average social contact following a reverse U-shape—rising through early adulthood, peaking in the mid-30s, and then declining, leaving older adults at risk for social isolation [13,30]. Finally, across cohorts, the average time spent alone in non-work activities is highest among the earliest cohorts (521 minutes for women, 458 for men), declines to a low of 188 minutes for women in the 1985 birth cohort and 212 for men in the 1979 birth cohort, and then rises again to 259 minutes for women and 313 for men in the 2007 cohort. As we will see, it is difficult to differentiate age and cohort effects with ATUS data and it is possible these cohort effects are more a reflection of life course processes than significant changes across cohorts in social isolation and social connectedness.

The simple averages illustrated in Fig 1 do not account for potential interrelationships between ages, periods, and cohorts. To identify their net contributions to time spent alone, we need to rely on a statistical model that allows each to be incorporated as predictors. This, however, poses a fundamental challenge due to the linear dependence of ages, periods, and cohorts – i.e., Age = Period – Cohort – which renders it impossible to include all three as predictors in a statistical model. Over the last 90 years, numerous approaches have been proposed to get around this problem [9,24–26]. The latest approach, described in detail below, accepts that linear effects of age, period, and cohort are not estimable and instead adopts an approach that [1] estimates parameters that define sets of linear age, period, and cohort effects that

are consistent with the data (referred to as a canonical solution line), [2] permits the specification of constraints that can bound linear age, period, and cohort effects, and [3] incorporates estimates of nonlinear deviations from the linear age, period, and cohort effects with the range of linear effects consistent with the constraints and the canonical solution line to identify net total age, period, and cohort effects [22].

Fig 2 illustrates the canonical solution line for linear age, period, and cohort effects on time spent alone in nonwork activities. The y-axis on the left indicates the range of linear age effects, the y-axis on the right indicates the range of linear cohort effects, and the x-axis indicates the range of linear period effects. The blue line, the canonical solution line, represents the sets of linear age, period, and cohort effects that are consistent with the data. For instance, in the case of women, one point on the line is a linear period effect of 0, which would come with a linear age effect of 5.15 and a linear cohort effect of −0.03.

To place bounds on the infinite set of linear effects represented by the canonical solution line, we turn to the well-established literature on age effects in social isolation that finds total social contact peaks among people in their mid-30s [13,30]. In other words, people are least likely to spend time alone in their mid-30s, as family and work responsibilities demand much of their time. We used a grid-search to identify minimum and maximum linear age effects that result in a

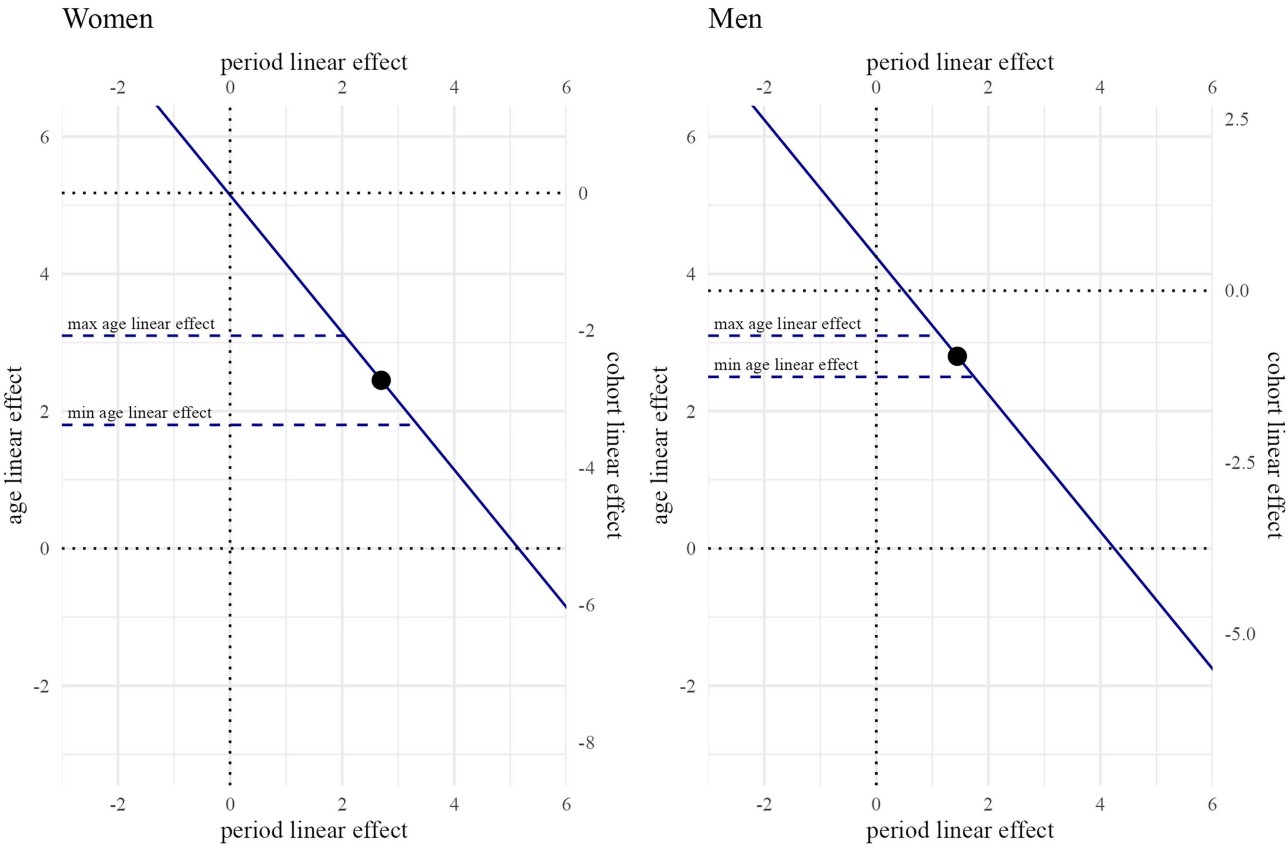

**Fig 2. Canonical solution line for linear age, period, and cohort effects for women and men.** Estimates for the line derived from an APC model described below that incorporates ATUS 2003−22 sample weights and adjusts for holidays and days of the week. The point along the line reflects the mid-point estimates for the linear effects used for the net total effects. The dashed lines demarcate the region of the lines based on maximum and minimum linear age effects consistent with the constraints described in the text. For women, the maximum linear age, period, and cohort effects are (3.1, 3.3, −2.1) and the minimum effects are (1.8, 2.0, −3.4). For men, the maximum linear age, period and cohort effects are (3.1, 1.7, −0.7) and the minimum effects are (2.5, 1.1, −1.3).

net total age effect with a minimum at age 35. We find a minimum linear age effect of 1.8 for women as well as a maximum linear age effect of 3.1 for women and 3.1 for men allow for the net total age effect to have a minimum at age 35. These bounds on the linear age effect demarcate a region of the canonical solution line that bounds the linear period effect to range from 2.0 to 3.3 for women and 1.1 to 1.7 for men as well as the linear cohort effect from −3.4 to −2.1 for women and −1.3 to −0.7 for men. The midpoint of this region of the canonical solution line, indicated by the point in Fig 2, has a linear age effect of 2.5 for women and 2.8 for men, a linear period effect of 2.7 for women and 1.4 for men, and a linear cohort effect of −2.7 for women and −1.0 for men. We use this range of linear effects and the midpoint to analyze the net total age, period, and cohort effects associated with minutes spent alone in nonwork activities.

Fig 3 illustrates our estimates for the net total effects of age, period, and cohort on minutes spent alone in nonwork activities. Beginning with the period effect, we see that the bounds on the linear effect result in tight range of estimates for the net total effect for both women and men. The midpoint estimates indicate an increase in time spent alone from 272 minutes for women and 273 minutes for men in 2003–314 minutes for women and 298 minutes for men in 2022. These effects are quite close to the simple averages and suggest that period effects are not meaningfully confounded by age and cohort effects. Furthermore, the net total effects also appear to show a slight uptick in the pattern in the mid 2010s.

Turning to the net total age effects, we observe a similar curvilinear shape as with the simple averages, but the effect sizes are meaningfully attenuated. The minimum occurs around age 35 (244 minutes for women, 233 minutes for men),

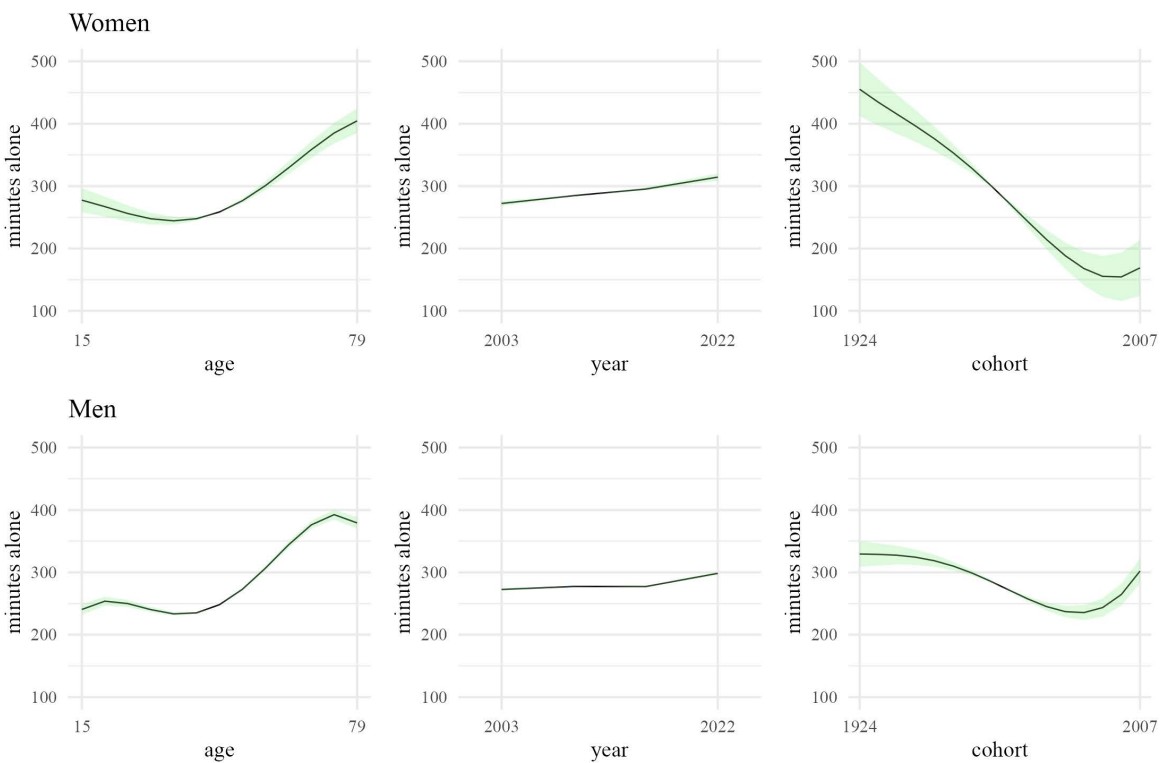

**Fig 3. APC model estimates of mean minutes spent alone in nonwork activities by age, period, and cohort for women and men.** Estimates combine linear and nonlinear estimates from a regression model based on applying the ATUS 2003−22 sample weights and adjusting for holidays and days of the week. Linear estimates reflect the range of effects consistent with the bounded region canonical solution line given in Fig 2. The black line with points represents the midpoint of the bounded linear effects. The green shading reflects the range of possible net total age, period, and cohort effects consistent with the bounded region of the canonical solution line.

and the maximum around age 79 (404 minutes) for women and around age 70 (392 minutes) for men. This attenuation is particularly pronounced for women, suggesting meaningful confounding by period and cohort effects.

For the net total cohort effects, a similar curvilinear shape as with the simple averages is observed for women. However, men's net cohort effect is much attenuated and relatively stable compared to the simple averages. For example, the gap between minimum and maximum is 94 minutes for the net cohort effect versus 246 minutes for the simple means (see Fig 1). Notably there is greater variation in the net total age and cohort effects that are consistent with the data than in the net total period effects. This is a likely reflection of the greater challenge in distinguishing age and cohort effects with ATUS data than in isolating period effects.

Our identification of net age, period, and cohort effects on minutes spent alone in nonwork activities permits a few observations. First, there does appear to have been a secular increase in the amount of time people spend alone over the last 20 years. This increase likely accelerated in the mid 2010s. Second, age or cohort effects are of much greater magnitude than period effects over this time period. Older adults and/or people born in early birth cohorts much more time alone in nonwork activities than middle-aged adults or people born in the late 1970s and early 1980s. Third, there is some evidence of an increase in the time spent alone among the most recent cohorts, but it is inconclusive given the structure of the ATUS data to disentangle age and cohort effects. Finally, gender differences in age and cohort effects are quite large, particularly for cohort effects and age effects below 20 and after 70.

### Sensitivity analyses

Although our main analysis adjusts for day of the week and holidays, we also conducted a stratified analysis by weekday versus weekend to explore potential differences. As shown in S1 Fig (supplementary material), overall effects are similar across both, but magnitudes differ: the age effect is more pronounced on weekdays, whereas period and cohort effects are more pronounced on weekends.

Rather than focusing on nonwork alone time, there are alternatives ways to operationalize free or discretionary time, such as leisure time. Leisure time is measured as activities such as Eating & Drinking (activity code 110101–119999 and 050202), Socializing, Relaxing, & Leisure (activity code 120101–129999 and 050201), and Sports, Exercise, & Recreation (activity code 130101–139999 and 050203) [31]. We conduct sensitivity analysis using leisure time spent alone (see S2 Fig in the supplementary material). The results are largely consistent with our main analyses, though the cohort effect for women is attenuated—from a threefold to a twofold difference—indicating that cohort differences in alone time are larger for non-work time than for leisure time. Overall, age, period, and cohort effects are similar in both contexts, suggesting the robustness of our findings. Therefore, we retain non-work time as our primary measure to align with the Surgeon General's report and other work in this area [1,10].

Given the COVID-19 pandemic has influenced people's social interactions [4], we conduct sensitivity analysis to examine whether our main findings are robust using pre-pandemic data (2003–2019). Our estimates of total effects are broadly similar when we conduct the analysis just using pre-pandemic data (2003–2019). The cohort estimates are slightly stronger for women and weaker for men than in the primary analysis. Clearly the pandemic had some effect, but it did not alter the general pattern of age, period, and cohort effects for men and women. We include a figure of total effects in the supplementary material (see S3 Fig).

### Discussion

Using 2003–2022 waves of ATUS, we leverage the latest breakthrough in the APC modeling to disentangle societal changes, age-specific effects, and generational shifts contributing to social isolation in the era of smartphones and social media, analyzed separately for women and men.

Starting with the period effect, the amount of time Americans spend alone has increased since 2003 for both women and men. This effect aligns with the U.S. Surgeon General's warning about "our epidemic of loneliness and isolation." [1]

Furthermore, the upward pattern appears to accelerate in the mid 2010s, suggesting that societal shifts during this period may have intensified social isolation. One pivotal shift may be the rise of smartphones and social media.

We find a U-shaped age effect in social isolation for women and men, consistent with findings in the literature [11–13]. The rise in social isolation in older age is often attributed to life events such as retirement, becoming empty nesters, and the loss of friends and family [11,14]. The socioemotional selectivity theory argues that older adults intentionally reduce social interactions that lack emotional meaning, which may result in fewer social interactions overall [13,30]. A notable gender difference is that the downward pattern after age 70 occurs for men, whereas it continues to rise for women. This may be attributed to women's higher rate of widowhood compared to men [32]. More importantly, the age effect is vastly larger than the period effect, with a difference of about 150 minutes between ages 35 and 79, compared to a 30-minute difference from 2003 to 2022. This suggests that, while there is a period effect indicating increased social isolation, its effect size pales in comparison to the age effect. Therefore, addressing social isolation among older adults may warrant a higher priority, especially given America's aging demographics [33].

The cohort effect resembles a mirror image of the age effect, suggesting a decrease in social isolation for newer cohorts, with the exception of the most recent cohorts. This is inconsistent with the literature, which typically reports either an increase or a stable cohort effect in social isolation [20,21]. We observed a substantial gender difference in cohort effects: older cohorts of women spend up to three times more time alone than younger cohorts, compared to less than twice as much for men. This suggests that changes in alone time across cohorts are more pronounced among women than among men. The cohort effect has the largest possible range, highlighting the challenge of disentangling cohort effects from age effects when using 20 years of cross-sectional data. This is because we only have older individuals from older cohorts and younger individuals from the most recent cohort. As a result, it becomes challenging to determine whether the cohort effect is simply a reflection of the age effect. Therefore, future studies should reevaluate the cohort effect in social isolation using data that spans more than 20 years.

It is important to keep in mind that an infinite number of potential combinations of age, period, and cohort effects are consistent with any given data and, ultimately, it requires theoretical assumptions and substantive knowledge to impose restrictions to help eliminate some of the infinite possibilities. We have imposed what we believe are modest assumptions, but this is also an avenue for future studies—identifying stronger or alternative assumptions that can further narrow the bounds on age, period, and cohort effects in time spent alone.

The observed increases in social isolation, both with age and over the 2003–2022 period, indicate a public health crisis and support the U.S. Surgeon General's warning about "our epidemic of loneliness and isolation." [1] Moreover, Putnam famously argues in "Bowling alone" that a decrease in face-to-face social interactions can lead to political polarization and distrust in public institutions [20].

## Supporting information

**S1 Fig. APC model estimates of mean minutes spent alone in nonwork activities by age, period, and cohort for weekdays and weekends.**
(DOCX)

**S2 Fig. APC model estimates of mean minutes spent alone in leisure activities by age, period, and cohort for women and men.**
(DOCX)

**S3 Fig. APC model estimates of mean minutes spent alone in nonwork activities by age, period, and cohort for women and men in the pre-pandemic period (2003–2019).**
(DOCX)

## Author contributions

**Conceptualization:** Siyun Peng.

**Formal analysis:** Shawn Bauldry.

**Methodology:** Shawn Bauldry.

**Visualization:** Shawn Bauldry.

**Writing – original draft:** Siyun Peng.

**Writing – review & editing:** Siyun Peng, Shawn Bauldry.

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
