## [Decision Letter · Decision Letter 0]

21 Jul 2025

Dear Dr. Peng,

publication criteria  and not, for example, on novelty or perceived impact.

We look forward to receiving your revised manuscript.

Kind regards,

Nicola Daniele Coniglio

Academic Editor

PLOS ONE

Additional Editor Comments:

Reviewers' comments:

Reviewer's Responses to Questions

**Comments to the Author**

1. Is the manuscript technically sound, and do the data support the conclusions?

Reviewer #1: Yes

Reviewer #2: Yes

2. Has the statistical analysis been performed appropriately and rigorously?

Reviewer #1: Yes

Reviewer #2: Yes

3. Have the authors made all data underlying the findings in their manuscript fully available?

Reviewer #1: Yes

Reviewer #2: Yes

4. Is the manuscript presented in an intelligible fashion and written in standard English?

Reviewer #1: Yes

Reviewer #2: Yes

Reviewer #1: Major Comments and Suggestions

1.1 I recommend extending the analysis to consider gender differences. The literature shows that trends in time-use activities vary significantly by gender, which may influence the period effect. Additionally, time-use patterns by age vary depending on the day of the week. I suggest conducting separate analyses for weekdays (Monday–Friday) and weekends/holidays (Saturday, Sunday, and public holidays). Discretionary time (and potentially time spent alone) is distributed very differently across these types of days, and it would be interesting to explore this heterogeneity.

1.2 Could the method and analysis be applied specifically to free or discretionary time spent alone? This would help isolate individual choice from more structured or obligatory time-use patterns (e.g., education). It could also strengthen the argument that societal shifts are driven by the rise of smartphones and social media, helping to control for other factors such as increased online learning post-pandemic.

1.3 How sensitive are your findings to the COVID period? Time spent alone changed substantially during the pandemic. Please also refer to the comment below regarding the use of sample weights during this period.

Minor Comments

1. In the abstract, the authors describe a 10% increase in the period trend over 20 years as "modest." While it may be modest relative to age and cohort effects, it is not necessarily modest in absolute terms and could be rephrased for clarity.

2. The presentation of the time-use data is appreciated, particularly for readers familiar with the dataset. However, more background information would benefit general readers who are less familiar with the data.

3. The ATUS definition of "time spent alone" introduces some ambiguity. The definition states: “Time spent alone includes activities in which ATUS respondents were asked who else was with them during an activity but did not report that others were present.” It is unclear whether “present” means physically present. For example, during online gaming or social media interactions, respondents may be physically alone (e.g., in a room) but socially engaged with others. If respondents report they were "with others" in such cases, the measure of physical solitude may be understated, potentially leading to an underestimation of the findings.

4. In the same vein, excluding time spent alone while working (particularly from home) could further understate the results, especially given the increasing prevalence of remote work since the pandemic.

5. How did you handle sample weights for 2020 (wt20), which are incompatible with the weights from other years (wt06)? Please clarify how you combined or adjusted them in the analysis.

6. The method for constructing cohort groups is not clearly described. Please provide more detail, similar to the explanation given for age and period groups.

7. In Equation (1), the subscripts L and NL presumably refer to linear and nonlinear components, respectively. Please clarify this explicitly in the text.

8. On page 7, change the reference “from (4)” to “from Equation (4)” to avoid confusion with citation references.

9. The use of the term “unconditional” to describe the results is ambiguous. Does it refer to simple averages or raw values without covariate adjustment? Please define the term more precisely.

10. Figure 2 is difficult to interpret. Consider adding numerical labels to the axes to improve readability. For example, the linear period effect of 0, linear age effect of 4.7, and cohort effect of 0.2 are mentioned in the text but are not clearly indicated in the figure. The same applies to the midpoint; please label these values directly in the graph to guide the reader.

Reviewer #2: This study examines the temporal effects of social isolation in the United States. Using data from the American Time Use Survey from 2003 to 2022, the authors employ an Age-Period-Cohort (APC) model to distinguish between the impacts of getting older, changes in society over time, and generational effects. Overall, this is a generally well-executed paper that could be strengthened with a few minor revisions.

2.1. In the "Analytic Method" section, the paper should more explicitly state its significant reliance on the work of Fosse and Winship. This is in terms of notation, model set up, approach, visualization, etc.

2.2. For greater clarity and consistency with the cited Fosse and Winship methodology, the authors should consistently use the term "effects" rather than using it interchangeably with "trends." In the context of an APC analysis, "effects" refer to the influence of the distinct underlying variables that age, period, and cohort represent, whereas "trends" are more appropriately used to describe the observable patterns in the data over time. In short, strike out the use of the word "trends" in the paper as this is an APC analysis. This wouldn't take much time at all and would greatly increase the clarity and scope of the paper.

2.3. The conclusion would be improved by adding a paragraph that emphasizes the tentative nature of the findings. An APC analysis is based on a rank deficient design matrix, meaning there are infinite possible solutions that fit the data; the selection of a particular solution is guided by theoretical assumptions rather than being solely data-driven, as the authors are no doubt aware.

**Do you want your identity to be public for this peer review?** For information about this choice, including consent withdrawal, please see our Privacy Policy

Reviewer #1: **Yes: ** Rezart Hoxhaj

Reviewer #2: No

---

## [Author Response · Author response to Decision Letter 1]

27 Aug 2025

Please see attached file for our response

---

## [Decision Letter · Decision Letter 1]

15 Sep 2025

In-person social isolation in the age of smartphones: examining age, period, cohort effects by gender.

PONE-D-25-31294R1

Dear Dr. Siyun Peng,

We’re pleased to inform you that your manuscript has been judged scientifically suitable for publication and will be formally accepted for publication once it meets all outstanding technical requirements.

I'm reporting some  final  suggestions by one of the referees that I invite you to consider during the final revision of the manuscript.

Kind regards,

Nicola Daniele Coniglio

Academic Editor

PLOS ONE

Additional Editor Comments (optional):

Reviewer #1:

Reviewer #2:

My only advice is to remove this sentence: "It is important to keep in mind that disentangling age, period, and cohort effects is not possible empirically due to the linear relationship between the three components that results in a rank deficient design matrix."

It's a weird statement as the authors do empirically separate the three effects, but with theoretical information supplied to identify a location on the solution line. I would instead start the paragraph with "It is important to keep in mind that an infinite..."

Another nitpick. The use of the word "trend" is a bit misleading as it is used in the paper. I strongly advise the authors to refer to "pattern" instead and scrub the use of the "trend" when using an APC model. With an APC model you are *fixing* the other two variables, so you are not looking at trends through (or with) time in any meaningful sense (i.e., observationally) but rather looking at apparent patterns of underlying effects.

Reviewers' comments:

Reviewer's Responses to Questions

**Comments to the Author**

Reviewer #1: (No Response)

Reviewer #2: All comments have been addressed

2. Is the manuscript technically sound, and do the data support the conclusions?

Reviewer #1: (No Response)

Reviewer #2: Yes

3. Has the statistical analysis been performed appropriately and rigorously?

Reviewer #1: (No Response)

Reviewer #2: Yes

4. Have the authors made all data underlying the findings in their manuscript fully available?

Reviewer #1: (No Response)

Reviewer #2: Yes

5. Is the manuscript presented in an intelligible fashion and written in standard English?

Reviewer #1: (No Response)

Reviewer #2: Yes

Reviewer #1: I think that the authors addressed my main concerns with this paper. In my view, it is ready to be published.

Reviewer #2: My only advice is to remove this sentence: "It is important to keep in mind that disentangling age, period, and cohort effects is not possible empirically due to the linear relationship between the three components that results in a rank deficient design matrix."

It's a weird statement as the authors do empirically separate the three effects, but with theoretical information supplied to identify a location on the solution line. I would instead start the paragraph with "It is important to keep in mind that an infinite..."

Another nitpick. The use of the word "trend" is a bit misleading as it is used in the paper. I strongly advise the authors to refer to "pattern" instead and scrub the use of the "trend" when using an APC model. With an APC model you are *fixing* the other two variables, so you are not looking at trends through (or with) time in any meaningful sense (i.e., observationally) but rather looking at apparent patterns of underlying effects.

**Do you want your identity to be public for this peer review?** For information about this choice, including consent withdrawal, please see our Privacy Policy

Reviewer #1: No

Reviewer #2: No

---

## [Editor Report · Acceptance letter]

PONE-D-25-31294R1

PLOS ONE

Dear Dr. Peng,

I'm pleased to inform you that your manuscript has been deemed suitable for publication in PLOS ONE. Congratulations! Your manuscript is now being handed over to our production team.

Kind regards,

on behalf of

Dr. Nicola Daniele Coniglio

Academic Editor

PLOS ONE